# Static Recrystallization Behavior of Low-Carbon Nb-V-Microalloyed Forging Steel

**Yang Zhao [1],\***[ID]**, Jiahao Zheng [2]**[ID]**, Liqing Chen [2] and Xianghua Liu [2,3]**

1   School of Materials Science and Engineering, Northeastern University, Shenyang 110819, China
2   State Key Laboratory of Rolling and Automation, Northeastern University, Shenyang 110819, China
3   Key Laboratory of Data Analytics and Optimization for Smart Industry, Ministry of Education, Northeastern University, Shenyang 110819, China
*   Correspondence: zhaoyang0323@smm.neu.edu.cn

**Abstract:** Static recrystallization is a method of tailoring the microstructure and mechanical properties of steels, which is important for microalloyed forging steels as the hot deformation process significantly affects their mechanical properties. In this paper, the static recrystallization behavior of a low-carbon Nb-V-microalloyed forging steel was investigated by double-pass hot compression tests at deformation temperature of 800–1100 °C and interruption time of 1–1000 s. The static recrystallization fractions were determined using the 2% offset method. The static recrystallization activation energy and the static recrystallization critical temperature (SRCT) of the experimental steel were determined. When the deformation temperature was higher than the SRCT, the recrystallization fraction curve conformed to the Avrami equation. When the deformation temperature was below the SRCT, the recrystallization curve appeared to plateau, which was caused by strain-induced precipitation. Before and after the plateau, the static recrystallization kinetics still obeyed the Avrami equation.

**Keywords:** Nb-V-microalloyed forging steel; static recrystallization; static recrystallization critical temperature; strain-induced precipitation

## 1. Introduction

Under the background of carbon neutrality, energy saving and emission reduction are gradually becoming a topic in the industry. The application of microalloyed forging steels (also known as non-quenched and tempered steels) is an important way to save energy and reduce emissions, as the heat treatment process (quenching and tempering) is cancelled in the production process [1–3]. Compared with quenched and tempered steels, microalloyed forging steels have comparable strength, but their toughness is inferior. Therefore, improving the toughness of microalloyed forging steels is the focus of their development. Optimizing the chemical composition and production process to promote the formation of acicular ferrite is an effective way to improve the toughness of microalloyed forging steels [4–6]. Miyamoto et al. [7] suggested that incoherent MnS + V(C,N) complex precipitate was the most preferential nucleation sites of acicular ferrite in V-microalloyed steels. Considering that second-phase precipitates can also increase the yield strength and tensile strength of microalloyed forging steel, it is important to control their precipitation behavior.

The production process of microalloyed forging steels consists mainly of controlled forging and controlled cooling. The hot deformation process of microalloyed forging steel contains several deformation passes. Static recrystallization and strain-induced precipitation of microalloyed steels can take place during the interruption time [8–11]. It has been found that the static recrystallization of vanadium-microalloyed steel is delayed compared with plain carbon steel [12]. The retardation of static recrystallization can be attributed to the solute drag effect of vanadium or the pinning effect of V(C,N) precipitates. Furthermore, fine V(C,N) can also refine the prior austenite grain. Cho et al. [13] studied the

static recrystallization behavior of a Nb-microalloyed steel and found that static recrystallization was significantly delayed compared with C-Mn steels, which can be accounted for by the solute drag effect of solid solution Nb. For vanadium-microalloyed steel, it was found that the static recrystallization fraction deviated from the Avrami equation when stain-induced precipitation began, which led to a plateau on the recrystallization time curves [14]. Zurob et al. [10] developed a physically based model to describe the interaction between static recrystallization and strain-induced precipitation in microalloyed steels, which can be used to predict the recrystallization fraction as a function of time. In addition, the model can be used to create recrystallization-time-temperature diagrams, as well as deformation-temperature diagrams. The occurrence of static recrystallization and strain-induced precipitation influences the microstructure and mechanical properties of microalloyed steels. Therefore, it is important to understand the static recrystallization behavior and strain-induced precipitation behavior. Nb-V-microalloyed forging steels are widely used in the production of shaft components for automobiles [15,16]. At present, there is little research on the static recrystallization behavior of Nb-V-microalloyed forging steels. To promote the development of Nb-V-microalloyed forging steels, it is necessary to investigate their static recrystallization behavior.

## 2. Materials and Methods

A low-carbon Nb-V-microalloyed forging steel was used as the experimental material, and its chemical composition is listed in Table 1. The experimental steel was smelted in an intermediate-frequency vacuum induction furnace and then hot rolled into a steel plate with a thickness of 12 mm. Thermal simulation specimens with dimensions of $\Phi 8$ mm $\times$ 15 mm were machined from the steel plate.

**Table 1.** Chemical composition of the low-carbon Nb-V-microalloyed forging steel (wt.%).

| C | Si | Mn | S | P | Cr | V | Nb |
|---|----|----|---|---|----|---|----|
| 0.24 | 0.30 | 1.99 | 0.050 | 0.008 | 0.51 | 0.08 | 0.07 |

To study the static recrystallization behavior of experimental steel, double-pass compression tests were carried out on an MMS-300 thermal simulation testing machine. To prevent oxidation of the specimens during the heating and deformation process, high-purity argon was injected as a protective gas prior to the thermal simulation tests. The specimens were heated to 1200 °C at a rate of 20 °C/s, held for 180 s, cooled to the deformation temperature at a rate of 10 °C/s, and then held for 30 s to eliminate the temperature gradient. The deformation temperatures were 800, 850, 900, 950, 1000, 1050, and 1100 °C. After holding at the deformation temperature for 30 s, the specimens were subjected to interrupted compression tests. The strain in both passes was 0.3, and the strain rate was 10 s$^{-1}$. The interruption time between passes was 1, 5, 10, 20, 50, 100, 400, and 1000 s. The experimental process of the double-pass compression test is shown in Figure 1.

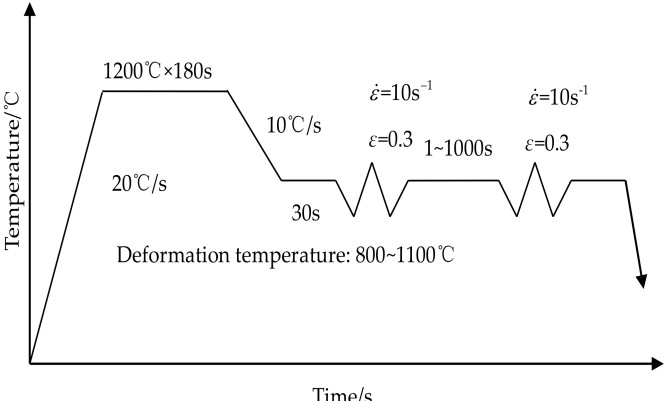

**Figure 1.** Schematic illustration of the double-pass compression tests.

To observe the microstructure of the experimental steel after different interruption times, some specimens were water-quenched to room temperature after the first pass and held for a certain time. The quenched specimens were etched in a solution of saturated picric acid and then observed by optical microscopy (OM) (Leica Q550IW, Weztlar, Germany). The foil and carbon replica samples were prepared to characterize the size, morphology, and distribution of precipitates, which were later observed using transmission electron microscopy (TEM) (Tecnai G$^2$ F20, Portland, OR, USA).

The 2% offset method was adopted to calculate the static recrystallization volume fraction of the experimental steel [17–19]. The static recrystallization fraction $X_s$ was measured as follows:

$$X_s = \frac{\sigma_m - \sigma_2}{\sigma_m - \sigma_1} \tag{1}$$

where $\sigma_m$ is the maximum flow stress of the first pass and $\sigma_1$ and $\sigma_2$ are the yield stresses in the first pass and second pass, respectively. The schematic diagram for determining the static recrystallization fraction by the 2% offset method is shown in Figure 2.

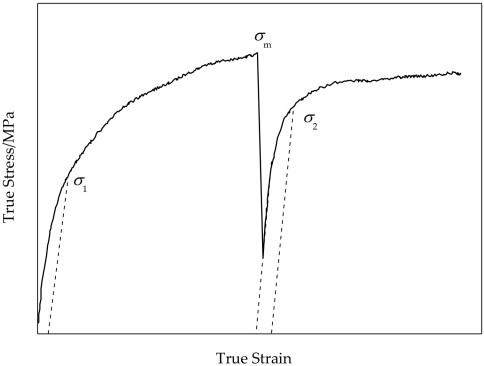

**Figure 2.** Schematic diagram of determining static recrystallization fraction by 2% offset method.

## 3. Results and Discussion

### 3.1. Static Recrystallization Fraction

The relationship between the static recrystallization fraction and interruption time at different deformation temperatures is shown in Figure 3. It can be seen that the recrystallization fraction increased with an increase in the deformation temperature at the same interruption time. At deformation temperatures of 1050 and 1100 °C, the recrystallization fraction curves were S-shaped and followed the Avrami equation; that is to say, the recrystallization fraction increased with the increasing interruption time at both deformation temperatures. Furthermore, the recrystallization fraction being up to one after a period of time at both deformation temperatures indicates that complete static recrystallization occurred. When the deformation temperature was at or below 1000 °C, a plateau appeared

on the recrystallization fraction curves. The appearance of the plateau means that static recrystallization was suppressed. It is suggested that strain-induced precipitation resulted in the plateau [20–23]. It should be noted that the recrystallization fraction still obeyed the Avrami equation before and after the plateau.

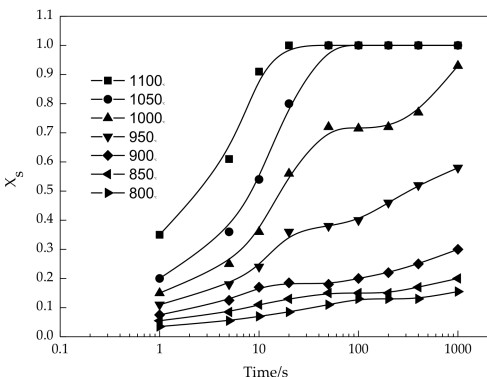

**Figure 3.** Relationship between static recrystallization fraction and interruption time.

*3.2. Static Recrystallization Activation Energy*

The relationship between the time $t_{0.5}$ corresponding to the recrystallization fraction reaching 0.5 and the static recrystallization activation energy can be expressed as [24,25]

$$t_{0.5} = A\varepsilon^p \dot{\varepsilon}^q D^s \exp\frac{Q}{RT} \tag{2}$$

where $Q$ is the static recrystallization activation energy, $R$ is the gas constant, $T$ is the absolute temperature, $\varepsilon$ is the true strain, $\dot{\varepsilon}$ is the strain rate, $D$ is the initial austenite grain size, and $A$, $p$, $q$, and $s$ are material-dependent constants.

By taking the natural logarithm of both sides of Equation (2), the following equation was obtained:

$$\ln t_{0.5} = \ln A + p \ln \varepsilon + q \ln \dot{\varepsilon} + s \ln D + \frac{Q}{RT} \tag{3}$$

A curve of $\ln t_{0.5}$ versus $1/T$ was drawn according to Equation (3), and it is shown in Figure 4. It can be seen that there was a linear relationship between $\ln t_{0.5}$ and $1/T$ when the deformation temperature was 1100, 1050, and 1000 °C. Based on the slope of the fitted line (the solid line in Figure 4), the static recrystallization activation energy was calculated to be 278.75 kJ/mol. It should be pointed out that strain-induced precipitation already occurred when the deformation temperature was 1000 °C. Because $t_{0.5}$ was less than the start time of strain-induced precipitation, the precipitation behavior had no effect on $t_{0.5}$, and therefore, the linear relationship between $\ln t_{0.5}$ and $1/T$ was still satisfied. When the deformation temperature was 950 °C, the onset of strain-induced precipitation was smaller than $t_{0.5}$, so $t_{0.5}$ was significantly delayed, and the linear relationship between $\ln t_{0.5}$ and $1/T$ was no longer achieved (as shown in Figure 4). At deformation temperatures of 900, 850, and 800 °C, $t_{0.5}$ could not be determined, as the recrystallization fraction did not reach 0.5 during the experimental time.

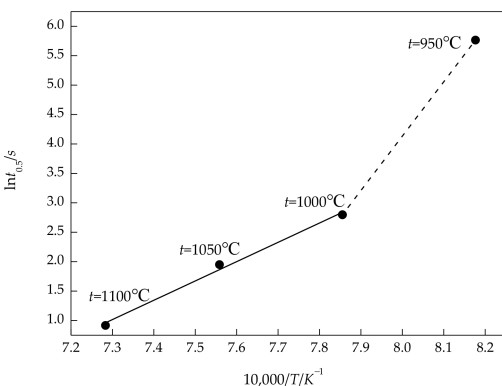

**Figure 4.** Relation between $\ln t_{0.5}$ and $1/T$ of experimental steel.

Medina and Quispe [25] studied the effect of chemical composition on the static recrystallization kinetics of low-alloyed steels and microalloyed steels using torsion tests and regressed the empirical relationship between the static recrystallization activation energy and chemical composition as shown in Equation (4):

$$Q(\text{J/mol}) = 148,636.8 - 71,981.3[C] + 56,537.6[Si] + 21,180[Mn] \\ + 121,243.3[Mo] + 64,469.6[V] + 109,731.9[Nb]^{0.15} \tag{4}$$

where each square bracket indicates the weight percentage of the displayed element. By substituting the chemical composition of the experimental steel into Equation (4), a calculated value of 269.27 kJ/mol could be obtained for the static recrystallization activation energy. Compared with the measured value of the static recrystallization activation energy, the calculated value was lower because the effect of Cr was not taken into account in Equation (4), while the experimental steel contained 0.51 wt.% Cr. From the above analysis, the static recrystallization activation energy value in this paper is reliable.

### 3.3. Static Recrystallization Microstructure Evolution

The austenite grain evolution of experimental steel with different interruption times at a deformation temperature of 900 °C is shown in Figure 5. When the interruption times were 1 and 5 s, there was almost no recrystallized grain (fine equiaxed grain), which indicates that the softening was mainly caused by static recovery. A few recrystallized grains appeared (as indicated by red arrows) when the interruption times were 10 and 20 s. The number of recrystallized grains increased further as the interruption time increased to 100 and 400 s, and the average austenite grain size decreased. Recrystallization is an essentially thermal diffusion process. As the interruption time increased, the diffusion of the elements continued, and therefore the recrystallization fraction gradually increased.

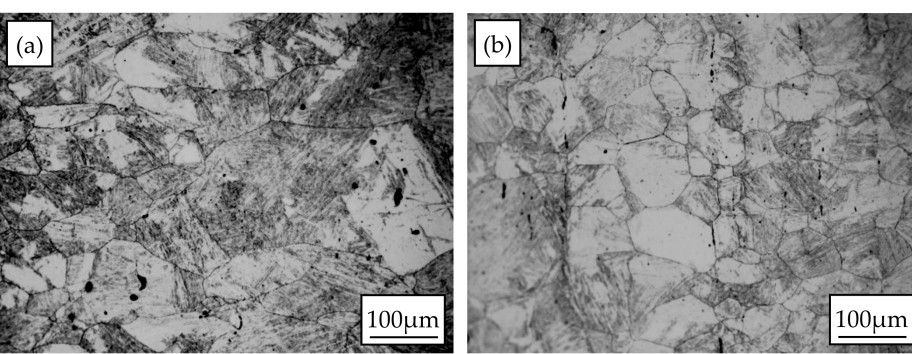

**Figure 5.** *Cont.*

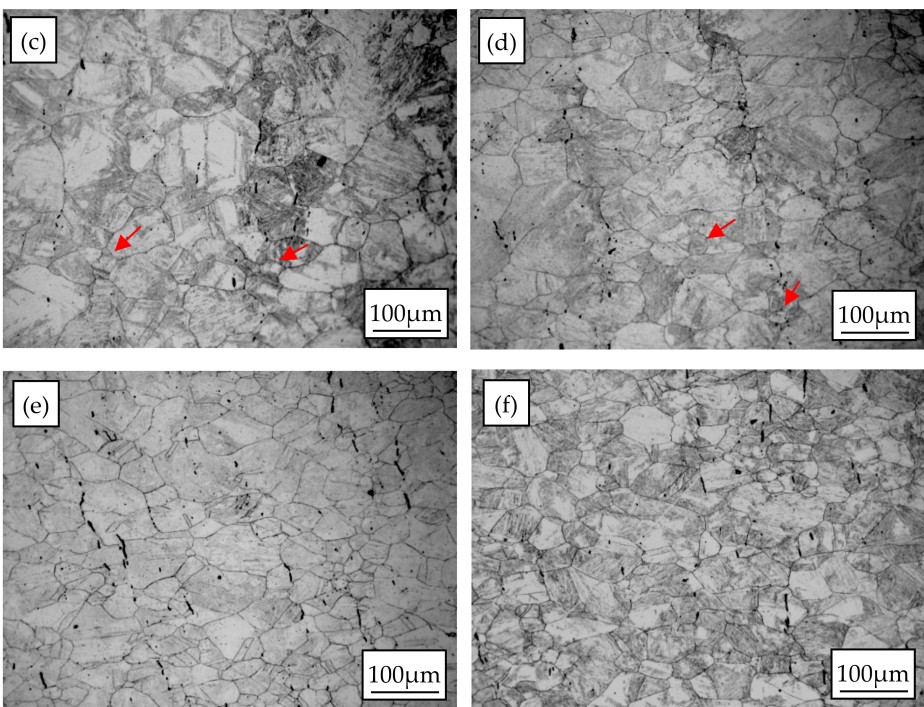

**Figure 5.** Microstructural evolution of experimental steel at deformation temperature of 900 °C and different interruption times: (**a**) 1 s, (**b**) 5 s, (**c**) 10 s, (**d**) 20 s, (**e**) 100 s, and (**f**) 400 s.

*3.4. Recrystallization-Precipitation-Time-Temperature Diagram*

A recrystallization-precipitation-time-temperature (RPTT) diagram is usually used to describe the interaction of static recrystallization with strain-induced precipitation [26–28]. The static recrystallization critical temperature (SRCT) of experimental steel should be determined prior to plotting the RPTT diagram. The SRCT is defined as the temperature at which the strain-induced precipitation starts to suppress static recrystallization [14]. This means that strain-induced precipitation only occurs when the deformation temperature is below the SRCT. As shown in Figure 6, the curve of $\ln Q$ and $10,000/T$ is plotted, and the abscissa value of the intersection of two straight lines is the SRCT [29]. The SRCT of the experimental steel was determined to be 1021 °C. It should be noted that $t_{0.5}$ at a deformation temperature of 900 °C was used to determine the SRCT, and $t_{0.5}$ was obtained by extrapolation using the recrystallization fraction curve in Figure 3. The SRCT of the experimental steel was similar to that of the Nb-microalloyed steel in Medina's study [29] and was higher than that of the V-microalloyed steel and Ti-microalloyed steel. This is because the SRCT mainly depends on the type and content of microalloyed elements in addition to the strain. It has been demonstrated that Nb-microalloyed steel has the highest SRCT among the three microalloyed steels [29].

The start and finish of the recrystallization fraction curve plateau correspond to the start and finish of the strain-induced precipitation, respectively. Therefore, the strain-induced precipitation start ($P_s$) and finish ($P_f$) times could be determined from Figure 3. In addition, Figure 3 was used to determine the temperatures and times corresponding to different recrystallization fractions, such as 0.1, 0.2, 0.3, 0.5, 0.7, and 0.95. By these means, the RPTT diagram (Figure 7) of the experimental steel was plotted. As can be seen from Figure 7, only static recrystallization occurred when the deformation temperature was above the SRCT. Both static recrystallization and strain-induced precipitation took place when the deformation temperature was below the SRCT. In addition, the recrystallization fraction did not vary between $P_s$ and $P_f$ and was denoted by a horizontal line. This demonstrates that below the SRCT, in order to obtain a specified recrystallization fraction, the strain-induced precipitation must be taken into account. Both the $P_s$ and $P_f$ curves had a C shape, and the nose temperature was about 900 °C. It can also be seen from Figure 7

that strain-induced precipitation did not occur when the strain was less than 0.1. The nose temperature of the $P_s$ curve in the present study was lower than that of the other steels with low Nb contents in Gómez's study [8]. In Gómez's study, the RPTT diagram was determined by torsion tests, and the mass fraction of Nb was 0.07%. The differences in the experimental method and chemical composition were responsible for the difference in nose temperature of the $P_s$ curves.

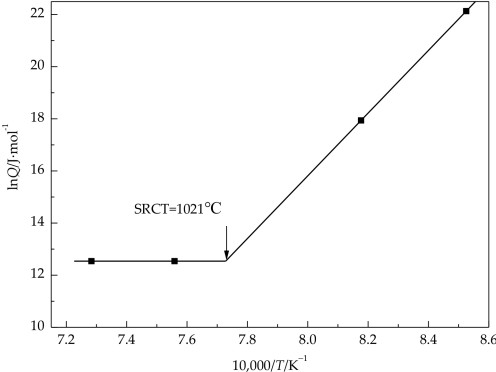

**Figure 6.** Plot of static recrystallization activation energy against the reciprocal of the absolute temperature.

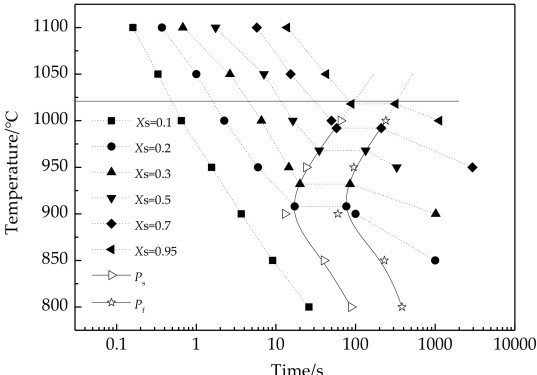

**Figure 7.** RPTT diagram of experimental steel.

The precipitates were observed here at a deformation temperature of 900 °C and at different interruption times to investigate the interaction between static recrystallization and strain-induced precipitation, as shown in Figure 8. Figure 8a shows the morphology of the strain-induced precipitates in the foil sample at the interruption time of 20 s. It can be determined from Figure 7 that the nose times of the $P_s$ and $P_f$ curves were 17 and 80 s, respectively. Strain-induced precipitation took place when the interruption time was 20 s. As shown in Figure 8a, the precipitate was located on dislocations, which indicates that the dislocations were the nucleation sites for the precipitates. Figure 8b,c shows the TEM images of the carbon replica samples with interruption times of 20 and 50 s, respectively. These two times were between the nose times of the $P_s$ and $P_f$ curves. As can be seen from Figure 8b,c, the number and size of the strain-induced precipitates increased over time. Furthermore, when the interruption time was 50 s, there were still small-sized precipitates, which indicates that the nucleation and growth of the precipitates were simultaneous during the strain-induced precipitation process. Figure 8d is the TEM photograph of the precipitates when the interruption time was 100 s. At this time, the strain-induced precipitation was complete. It can be seen from Figure 8d that coarsening of the precipitates occurred with increasing interruption times after the end of strain-induced precipitation.

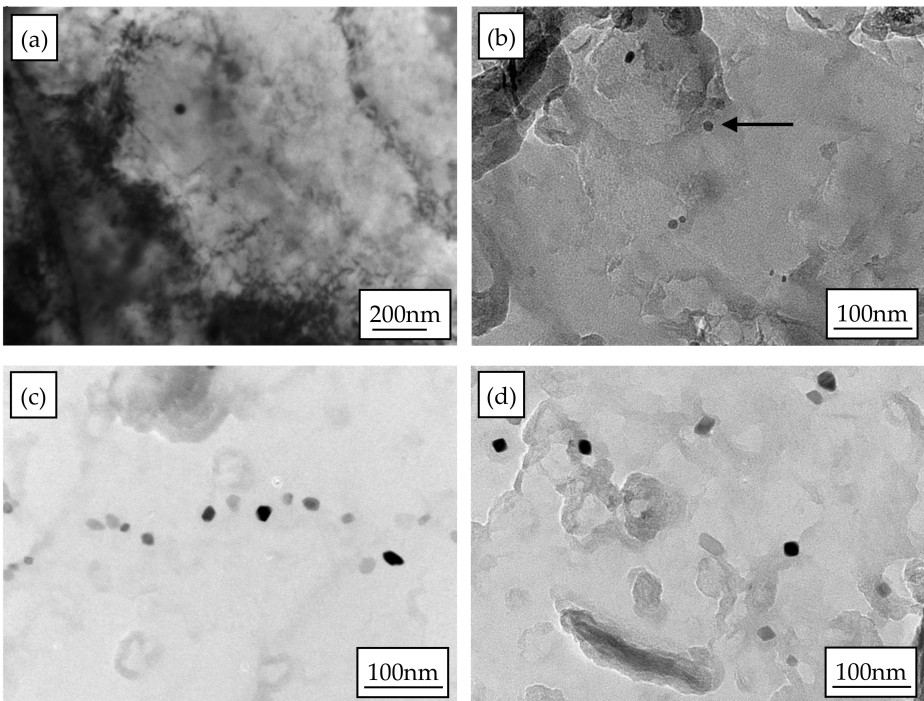

**Figure 8.** TEM images of strain-induced precipitates in experimental steel with different interruption times at deformation temperature of 900 °C: (**a**) 20 s, foil sample, (**b**) 20 s, carbon replica sample, (**c**) 50 s, carbon replica sample, and (**d**) 100 s, carbon replica sample.

Prior austenite grain boundaries and dislocations are the preferred nucleation sites of strain-induced precipitates [30,31]. It has been demonstrated that the essence of static recrystallization is to eliminate the defects in the materials [32]. If static recrystallization occurs before strain-induced precipitation, then it reduces the number of defects in the materials (i.e., it reduces the number of nucleation sites and therefore delays the onset of strain-induced precipitation). If strain-induced precipitation occurs during static recrystallization, recrystallization is inhibited because the pinning force by the fine precipitates is larger than the driving force for recrystallization [33,34]. As the interruption time increases, the size of the precipitates increases, which results in a reduction in pinning force. When the pinning force is less than the driving force for recrystallization, static recrystallization takes place again. This is the reason for the plateau in the recrystallization fraction curves.

## 4. Conclusions

(1) The static recrystallization activation energy of experimental steel was calculated to be 278.75 kJ/mol. When the strain was 0.3, the static recrystallization critical temperature (SRCT) was 1021 °C.

(2) The static recrystallization kinetics of the experimental steel followed the Avrami equation when the deformation temperature was higher than the SRCT. As the deformation temperature fell below the SRCT, strain-induced precipitation occurred, which led to a plateau in the recrystallization fraction curve. The recrystallization fraction remained consistent with the Avrami equation before and after the plateau.

**Author Contributions:** Methodology, Y.Z.; formal analysis, Y.Z.; investigation, Y.Z. and J.Z.; data curation, Y.Z.; writing—original draft preparation, Y.Z. and J.Z.; writing—review and editing, L.C. and X.L.; supervision, L.C. and X.L.; funding acquisition, Y.Z. All authors have read and agreed to the published version of the manuscript.

**Funding:** This research was financially supported by Fundamental Research Funds for the Central Universities (No. N2002024) and the 111 Project of China (B16009).

**Data Availability Statement:** Not applicable.

**Acknowledgments:** The authors acknowledge Weina Zhang for helping with the TEM experiments.

**Conflicts of Interest:** The authors declare no conflict of interest.

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
