# Peer review of "Static Recrystallization Behavior of Low-Carbon Nb-V-Microalloyed Forging Steel"

_metals, doi:10.3390/met12101745_

Round 1

Reviewer 1 Report

This paper presents the results of a systematic study of the recrystallization behavior of Nb-V microalloyed forging steel according to recrystallization temperature and time. 

It can be published after a few minor revision below.

1)     It is necessary to explain how the recrystallized grains were classified and marked in Figure 5.

2)     In Fig. 7, the nose of the C curve is between 0.1 and 0.3 recrystallization fractions. You said that precipitation does not occur when the recrystallization fraction is less than 0.2, but it is necessary to check whether this is an accurate value. For example, can it be 0.15 or 0.25? Can you add a line for Xs=0.2, in Fig. 7?

Author Response

Dear Reviewer:  

    According to your comments, the following revisions were made. 

(1) The morphological characteristics of the recrystallized grains are described in the revised manuscript.

(2) The line for Xs=0.2 is added in Figure 7. The recrystallization fraction corresponding to the nose temperature of the Ps curve is determined to be 0.17. However, it is found that when the deformation temperature is 800 ℃ and 850 ℃, the corresponding recrystallization fraction is less than 0.17 when strain-induced precipitation occurs, so the third conclusion is not valid. The third conclusion was deleted. 

Reviewer 2 Report

The authors report the investigation of the static recrystallization of a low carbon Nb-V microalloyed forging steel. The topic is interesting and the deserve to be investigated. The used methodology is appropriate. However, the authors investigate only one sample at one composition.

This is not enought to define the behaviour of this kind of alloys. An investigation to a set of different samples can support all hypothesys,

At thid stage, the manuscript cannot be accepted for the publication.

Author Response

(The authors gave the same response as above.)

Round 2

Reviewer 2 Report

The authors replied properly to the requests of reviewers.

The manuscript can be published.